# Dynamics-Regulated Kinematic Policy for Egocentric Pose Estimation

**Zhengyi Luo**[1]    **Ryo Hachiuma** [2] *    **Ye Yuan**[1]    **Kris Kitani**[1]

[1] Carnegie Mellon University    [2] Keio University

https://zhengyiluo.github.io/projects/kin_poly/

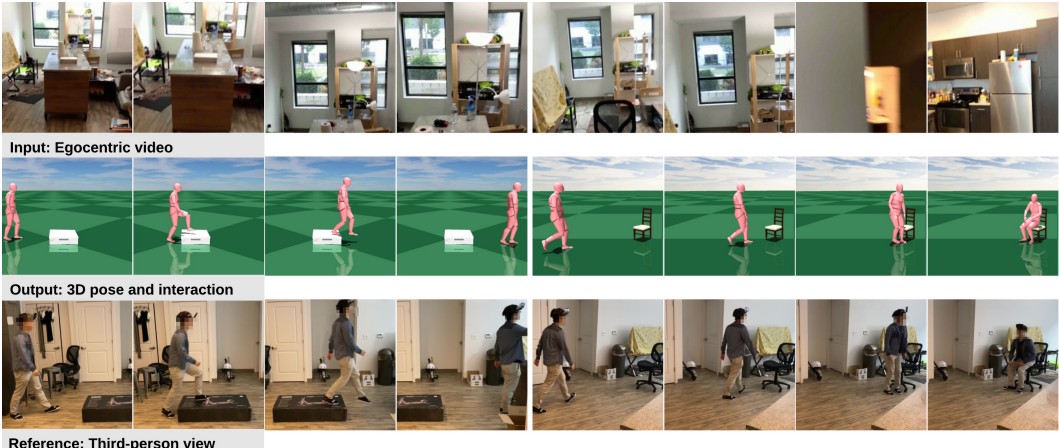

Figure 1: From egocentric videos, we infer physically-plausible 3D human pose and human-object interaction.

## Abstract

We propose a method for object-aware 3D egocentric pose estimation that tightly integrates kinematics modeling, dynamics modeling, and scene object information. Unlike prior kinematics or dynamics-based approaches where the two components are used disjointly, we synergize the two approaches via *dynamics-regulated training*. At each timestep, a kinematic model is used to provide a target pose using video evidence and simulation state. Then, a prelearned dynamics model attempts to mimic the kinematic pose in a physics simulator. By comparing the pose instructed by the kinematic model against the pose generated by the dynamics model, we can use their misalignment to further improve the kinematic model. By factoring in the 6DoF pose of objects (e.g., chairs, boxes) in the scene, we demonstrate for the first time, the ability to estimate physically-plausible 3D human-object interactions using a single wearable camera. We evaluate our egocentric pose estimation method in both controlled laboratory settings and real-world scenarios.

## 1   Introduction

From a video captured by a single head-mounted wearable camera (*e.g*., smartglasses, action camera, body camera), we aim to infer the wearer's global 3D full-body pose and interaction with objects in the scene, as illustrated in Fig. 1. This is important for applications like virtual and augmented reality, sports analysis, and wearable medical monitoring, where third-person views are often unavailable and proprioception algorithms are needed for understanding the actions of the camera wearer. However, this task is challenging since the wearer's body is often unseen from a first-person view and the

---

*Work done at Carnegie Mellon University.

35th Conference on Neural Information Processing Systems (NeurIPS 2021), virtual.

body motion needs to be inferred solely based on the videos captured by the *front-facing* camera. Furthermore, egocentric videos usually capture the camera wearer interacting with objects in the scene, which adds additional complexity in recovering a pose sequence that agrees with the scene context. Despite these challenges, we show that it is possible to infer accurate human motion and human-object interaction from a single head-worn front-facing camera.

Egocentric pose estimation can be solved using two different paradigms: (1) a kinematics perspective and (2) a dynamics perspective. *Kinematics-based approaches* study motion without regard to the underlying forces (*e.g.*, gravity, joint torque) and cannot faithfully emulate human-object interaction without modeling proper contact and forces. They can achieve accurate pose estimates by directly outputting joint angles but can also produce results that violate physical constraints (*e.g.* foot skating and ground penetration). *Dynamics-based approaches*, or *physics-based approaches*, study motions that result from forces. They map directly from visual input to control signals of a human proxy (humanoid) inside a physics simulator and recover 3D poses through simulation. These approaches have the crucial advantage that they output physically-plausible human motion and human-object interaction (*i.e.*, pushing an object will move it according to the rules of physics). However, since no joint torque is captured in human motion datasets, physics-based humanoid controllers are hard to learn, generalize poorly, and are actively being researched [36, 57, 51, 52].

In this work, we argue that a *hybrid* approach merging the kinematics and dynamics perspectives is needed. Leveraging a large human motion database [29], we learn a task-agnostic dynamics-based humanoid controller to mimic broad human behaviors, ranging from every day motion to dancing and kickboxing. The controller is *general-purpose* and can be viewed as providing low-level motor skills of a human. After the controller is learned, we train an object-aware kinematic policy to specify the target poses for the controller to mimic. One approach is to let the kinematic model produce target motion *only* based on the visual input [58, 51, 56]. This approach uses the physics simulation as a post-processing step: the kinematic model computes the target motion separately from the simulation and may output unreasonable target poses. We propose to synchronize the two aspects by designing a kinematic policy that guides the controller and receives timely feedback through comparing its target pose and the resulting simulation state. Our model thus serves as a high-level motion planning module that adapts intelligently based on the current simulation state. In addition, since our kinematic policy only outputs poses and does not model joint torque, it can receive direct supervision from motion capture (MoCap) data. While poses from MoCap can provide an initial-guess of target motion, our model can search for better solutions through trial and error. This learning process, dubbed *dynamics-regulated training*, jointly optimizes our model via supervised learning and reinforcement learning, and significantly improves its robustness to real-world use cases.

In summary, our contributions are as follows: (1) we are the first to tackle the challenging task of estimating physically-plausible 3D poses and human-object interactions from a single front-facing camera; (2) we learn a general-purpose humanoid controller from a large MoCap dataset and can perform a broad range of motions inside a physics simulation; (3) we propose a dynamics-regulated training procedure that synergizes kinematics, dynamics, and scene context for egocentric vision; (4) experiments on a controlled motion capture laboratory dataset and a real-world dataset demonstrate that our model outperforms other state-of-the-art methods on pose-based and physics-based metrics, while generalizing to videos taken in real-world scenarios.

## 2   Related Work

**Third-person human pose estimation.** The task of estimating the 3D human pose (and sometimes shape) from *third-person* video is a popular research area in the vision community [2, 21, 24, 14, 40, 59, 10, 34, 12, 31, 23, 28, 11, 54, 58], with methods aiming to recover 3D joint positions [25, 46, 14, 34], 3D joint angles with respect to a parametric human model [2, 21, 23, 28], and dense body parts [11]. Notice that all these methods are purely kinematic and disregard physical reasoning. They also do not recover the global 3D root position and are evaluated by zeroing out the body center (root-relative). A smaller number of works factor in human dynamics [42, 43, 38, 49, 4] through postprocessing, physics-based trajectory optimization, or using a differentiable physics model. These approaches can produce physically-plausible human motion, but since they do not utilize a physics simulator and does not model contact, they can not faithfully model human-object interaction. SimPoE [58], a recent work on third-person pose estimation using simulated character control, is most related to ours, but 1) trains a single and dataset-specific humanoid controller per dataset; 2) designs the kinematic model to be independent from simulation states.

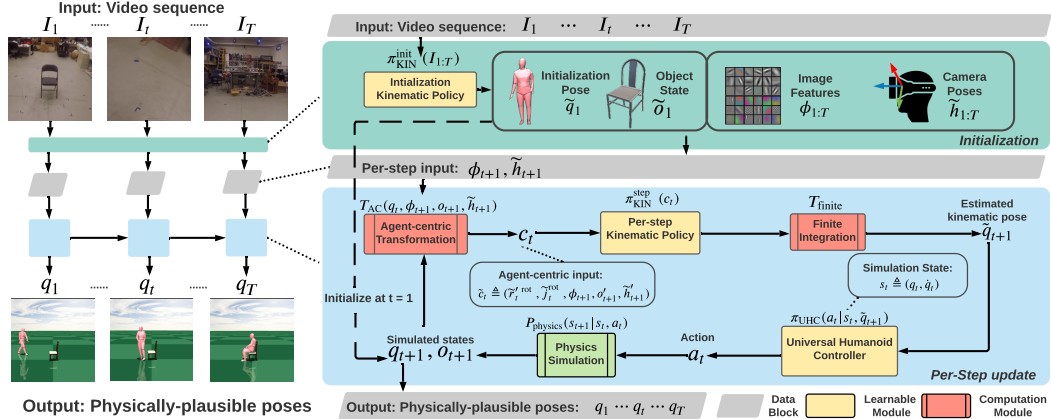

Figure 2: Overview of our dynamics-regulated kinematic policy. Given an egocentirc video $\boldsymbol{I}_{1:T}$, our initialization module $\boldsymbol{\pi}_{\mathrm{KIN}}^{\mathrm{init}}(\boldsymbol{I}_{1:T})$ computes the first-frame object state $\widetilde{\boldsymbol{o}}_1$, human pose $\widetilde{\boldsymbol{q}}_1$, camera poses $\widetilde{\boldsymbol{h}}_{1:T}$, and image features $\boldsymbol{\phi}_{1:T}$. The object state $\widetilde{\boldsymbol{o}}_1$ and human pose $\widetilde{\boldsymbol{q}}_1$ are used to initialize the phsycis simulation. At each time step, we roll out our per-step kinematic policy $\boldsymbol{\pi}_{\mathrm{KIN}}^{\mathrm{step}}$ together with the Universal Humanoid Controller to output physically-plausible pose $\boldsymbol{q}_t$ inside a physics simulator.

**Egocentric human pose estimation.** Compared to third-person human pose estimation, there are only a handful of attempts at estimating 3D full body poses from egocentric videos due to the ill-posed nature of this task. Most existing methods still assume partial visibility of body parts in the image [48, 39, 53], often through a downward-facing camera. Among works where the human body is mostly not observable [20, 55, 56, 33], Jiang *et al*. [20] use a kinematics-based approach where they construct a motion graph from the training data and recover the pose sequence by solving the optimal pose path. Ng *et al*. [33] focus on modeling person-to-person interactions from egocentric videos and inferring the wearer's pose conditioning on the other person's pose. The works most related to ours are [55, 56, 19] which use dynamics-based approaches and map visual inputs to control signals to perform physically-plausible human motion inside a physics simulation. They show impressive results on a set of noninteractive locomotion tasks, but also observe large errors in absolute 3D position tracking–mapping directly from the visual inputs to control signals is a noisy process and prone to error accumulation. In comparison, our work jointly models kinematics and dynamics, and estimates a wider range of human motion and human-object interactions while improving absolute 3D position tracking. To the best of our knowledge, we are the first approach to estimate the 3D human poses from egocentric video while factoring in human-object interactions.

**Humanoid control inside physics simulation.** Our work is also connected to controlling humanoids to mimic reference motion [36, 37, 5, 57, 52, 15, 6] and interact with objects [5, 30] inside a physics simulator. The core motivation of these works is to learn the necessary dynamics to imitate or generate human motion in a physics simulation. Deep RL has been the predominant approach in this line of work since physics simulators are typically not end-to-end differetiable. Goal-oriented methods [5, 30, 1] does not involve motion imitation and are evaluated on task completion (moving an object, sitting on a chair, moving based on user-input *etc*.). Consequently, these frameworks only need to master a subset of possible motions for task completion. People, on the other hand, have a variety of ways to perform actions, and our agent has to follow the trajectory predefined by egocentric videos. Motion imitation methods [36, 37, 52, 57, 52] aim to control characters to mimic a sequence of reference motion, but have been limited to performing a single clip [36, 37, 52, 57] or high-quality MoCap [52] motion (and requires fine-tuning to generalize to other motion generators). In contrast, our dynamics controller is general and can be used to perform everyday motion and human-object interactions estimated by a kinematic motion estimator without task-specific fine-tuning.

## 3    Method

The problem of egocentric pose estimation can be formulated as follows: from a wearable camera footage $\boldsymbol{I}_{1:T}$, we want to recover the wearer's ground truth global 3D poses $\widehat{\boldsymbol{q}}_{1:T}$. Each pose $\widehat{\boldsymbol{q}}_t \triangleq (\widehat{\boldsymbol{r}}_t^{\mathrm{pos}}, \widehat{\boldsymbol{r}}_t^{\mathrm{rot}}, \widehat{\boldsymbol{j}}_t^{\mathrm{rot}})$ consists of the root position $\widehat{\boldsymbol{r}}_t^{\mathrm{pos}}$, root orientation $\widehat{\boldsymbol{r}}_t^{\mathrm{rot}}$, and body joint angles $\widehat{\boldsymbol{j}}_t^{\mathrm{rot}}$ of the human model. Here we adopt the popular SMPL [27] human model and the humanoid we

use in physics simulation is created from the kinematic structure and mean body shape defined by SMPL. Our framework first learns a Universal Humanoid Controller (UHC) from a large MoCap dataset (Sec. 3.1). The learned UHC can be viewed as providing the lower level muscle skills of a real human, trained by mimicking thousands of human motion sequences. Using the trained UHC, we learn our kinematic policy (Sec. 3.2) through dynamics-regulated training (Sec. 3.3). At the test time, the kinematic policy provides per-step target motion to the UHC, forming a closed-loop system that operates inside the physics simulation to control a humanoid. The result of the UHC and physics simulation is then used as input to the kinematic policy to produce the next-frame target motion, as depicted in Fig. 2. As a notation convention, we use $\widetilde{\cdot}$ to denote kinematic quantities (obtained without using physics simulation), $\widehat{\cdot}$ to denote ground truth quantities, and normal symbols without accents to denote quantities from the physics simulation.

## 3.1 Dynamics Model - Universal Humanoid Controller (UHC)

To learn a task-agnostic dynamics model that can be tightly integrated with a kinematic model, we design our controller's state space to only rely on the current simulated pose $q_t$ and target posed $\widehat{q}_{t+1}$ and remove all phase or sequence-level information found in prior arts [57, 36, 37]. This design allows us to train on an order of magnitude larger dataset of human motion [29] with only pose information and significantly improve our models' ability to mimic diverse and unseen motions. Formally, we model controlling a humanoid to follow a reference motion $\widehat{q}_{1:T}$ as a Markov Decision Process (MDP) defined as a tuple $\mathcal{M} = \langle S, A, P_{\text{physics}}, R, \gamma \rangle$ of states, actions, transition dynamics, reward function, and a discount factor. The state $S$, reward $R$, and transition dynamics $P_{\text{physics}}$ are provided by the physics simulator, while action $A$ is computed by the policy $\pi_{\text{UHC}}$. At each timestep $t$, the agent in state $s_t$ takes an action sampled from the policy $\pi_{\text{UHC}}(a_t|s_t, \widehat{q}_{t+1})$ while the environment generates the next state $s_{t+1}$ and reward $r_t$. We employ Proximal Policy Optimization (PPO) [41] to find the optimal policy $\pi_{\text{UHC}}^*$ that maximizes the expected discounted return $\mathbb{E}[\sum_{t=1}^{T} \gamma^{t-1} r_t]$.

**State.** The state $s_t \triangleq (q_t, \dot{q}_t)$ of the humanoid contains the character's current pose $q_t$ and joint velocity $\dot{q}_t$. Here, the state $s_t$ encapsulates the humanoid's full physical state at time step $t$. It only includes information about the current frame $(q_t, \dot{q}_t)$ and does not include any extra information, enabling our learned controller to be guided by a target pose only.

**Action.** The action $a_t$ specifies the target joint angles for the proportional derivative (PD) controller [45] at each degree of freedom (DoF) of the humanoid joints except for the root (pelvis). We use the residual action representation [15]: $q_t^d = \widehat{q}_{t+1} + a_t$, where $q_t^d$ is the final PD target, $a_t$ is the output of the control policy $\pi_{\text{UHC}}$, and $\widehat{q}_{t+1}$ is the target pose. The torque to be applied at joint $i$ is: $\tau^i = k^p \circ (q_t^d - q_t) - k^d \circ \dot{q}_t$ where $k^p$ and $k^d$ are manually specified gains and $\circ$ is the element-wise multiplication. As observed in prior work [57, 58], allowing the policy to apply external residual forces $\eta_t$ to the root helps stabilizing the humanoid, so our final action is $a_t \triangleq (\Delta \widetilde{q}_t^d, \eta_t)$.

**Policy.** The policy $\pi_{\text{UHC}}(a_t|s_t, \widehat{q}_{t+1})$ is represented by a Gaussian distribution with a fixed diagonal covariance matrix $\Sigma$. We first use a feature extraction layer $D_{\text{diff}}(\widehat{q}_{t+1}, q_t)$ to compute the root and joint offset between the simulated pose and target pose. All features are then translated to a root-relative coordinate system using an agent-centic transform $T_{\text{AC}}$ to make our policy orientation-invariant. We use a Multi-layer Perceptron (MLP) as our policy network to map the augmented state $T_{\text{AC}}\left(q_t, \dot{q}_t, \widehat{q}_{t+1}, D_{\text{diff}}(\widehat{q}_{t+1}, q_t)\right)$ to the predicted action $a_t$.

**Reward function.** For UHC, the reward function is designed to encourage the simulated pose $q_t$ to better match the target pose $\widehat{q}_{t+1}$. Since we share a similar objective (mimic target motion), our reward is similar to Residual Force Control [57].

**Training procedure.** We train our controller on the AMASS [29] dataset, which contains 11505 high-quality MoCap sequences with 4000k frame of poses (after removing sequences involving human-object interaction like running on a treadmill). At the beginning of each episode, a random fixed length sequence (300 frames) is sampled from the dataset for training. While prior works [52, 51] uses more complex motion clustering techniques to sample motions, we devise a simple yet empirically effective sampling technique by inducing a probability distribution based on the value function. For each pose frame $\widehat{q}_j$ in the dataset, we first compute an initialization state $s_1^j$: $s_1^j \triangleq (\widehat{q}_j, 0)$, and then score it using the value function to access how well the policy can mimic the sequence $\widehat{q}_{j:T}$ starting from this pose: $V(s_1^j, \widehat{q}_{j+1}) = v_j$. Intuitively, the higher $v_j$ is, the more confident our policy is in mimicking this sequence, and the less often we should pick this frame. The

probability of choosing frame $j$, comparing against all frames $J$ in the AMASS dataset, is then $P(\widehat{q}_j) = \frac{\exp(-v_j/\tau)}{\sum_i^J \exp(-v_i/\tau)}$ where $\tau$ is the sampling temperature. More implementation details about the reward, training and evaluation of UHC can be found in Appendix C.

## 3.2 Kinematic Model – Object-aware Kinematic Policy

To leverage the power of our learned UHC, we design an auto-regressive and object-aware kinematic policy to generate per-frame target motion from egocentric inputs. We synchronize the state space of our kinematic policy and UHC such that the policy can be learned with or without physics simulation. When trained without physics simulation, the model is purely kinematic and can be optimized via supervised learning; when trained with a physics simulation, the model can be optimized through a combination of supervised learning and reinforcement learning. The latter procedure, coined *dynamics-regulated training*, enables our model to distill human dynamics information learned from large-scale MoCap data into the kinematic model and learns a policy more robust to convariate shifts. In this section, we will describe the architecture of the policy itself and the training through supervised learning (without physics simulation).

**Scene context modelling and initialization.** To serve as a high-level target motion estimator for egocentric videos with potential human-object interaction, our kinematic policy needs to be object-aware and grounded with visual input. To this end, given an input image sequence $I_{1:T}$, we compute the initial object states $\widetilde{o}_1$, image features $\phi_{1:T}$, and camera trajectory $\widetilde{h}_{1:T}$ as inputs to our system. The object states, $\widetilde{o}_t \triangleq (\widetilde{o}_t^{cls}, \widetilde{o}_t^{\text{pos}}, \widetilde{o}_t^{\text{rot}})$, is modeled as a vector concatenation of the main object-of-interest's class $\widetilde{o}_t^{cls}$, 3D position $\widetilde{o}_t^{\text{pos}}$, and rotation $\widetilde{o}_t^{\text{rot}}$. $\widetilde{o}_t$ is computed using an off-the-shelf object detector and pose estimator [17]. When there are no objects in the current scene (for walking and running *etc.*), the object states vector is set to zero. The image features $\phi_{1:T}$ contains crucial information about the wearer's movement and is computed using an optical flow extractor [44] and ResNet [13]. Since visual input can be noisy and modern smartglasses and bodycams are often equipped with built-in SLAM or Visual Inertial Odometry (VIO) [50, 9] capabilities, we utilize this additional data modalitiy and compute the 6DoF camera pose from input Images. Using an off-the-shelf VIO method [16], we extracts camera trajectory as: $\widetilde{h}_t \triangleq (\widetilde{h}_t^{\text{pos}}, \widetilde{h}_t^{\text{rot}})$ (position $\widetilde{h}_t^{\text{pos}}$ and orientation $\widetilde{h}_t^{\text{rot}}$). Notice that the camera trajectory $\widetilde{h}_t$ information is crucial and significantly improve the performance of our framework as shown in our ablation studies (Sec. 4.2).

To provide our UHC with a plausible initial state for simulation, we estimate $\widetilde{q}_1$ from the scene context features $\phi_{1:T}$, $\widetilde{o}_{1:T}$, and $\widetilde{h}_{1:T}$. We use an Gated Recurrent Unit (GRU) [7] based network to regress the initial agent pose $\widetilde{q}_1$. Combining the above procedures, we obtain the context modelling and initialization model $\pi_{\text{KIN}}^{\text{init}}$: $[\widetilde{q}_1, \phi_{1:T}, \widetilde{o}_1, \widetilde{h}_{1:T}] = \pi_{\text{KIN}}^{\text{init}}(I_{1:T})$. Notice that to constrain the ill-posed problem of egocentric pose estimation, we assume known object category, rough size, and potential mode of interaction. We use these knowledge as a prior for our per-step model for pose estimation.

**Training kinematic policy via supervised learning.** After initialization, we use the estimated first-frame object pose $\widetilde{o}_t$ and human pose $\widetilde{q}_t$ to *initialize* the physics simulation. All subsequent object movements are a result of human-object interation and simulation. At each time step, we use a per-step model $\pi_{\text{KIN}}^{\text{step}}$ to compute the next frame pose based on the next frame observations: we obtain an egocentric input vector $\widetilde{c}_t$ through the agent-centric transformation function $\widetilde{c}_t = T_{\text{AC}}(\widetilde{q}_t, \phi_{t+1}, \widetilde{o_{t+1}}, \widetilde{h}_{t+1})$ where $\widetilde{c}_t \triangleq (\widetilde{r}_t^{\prime rot}, \widetilde{j}_t^{\text{rot}}, \phi_{t+1}, \widetilde{o}'_{t+1}, \widetilde{h}'_{t+1})$ contains the current agent-centric root orientation $\widetilde{r}_t^{\prime rot}$, joint angles $\widetilde{j}_t^{\text{rot}}$, image feature for next frame $\phi_{t+1}$, object state $\widetilde{o}'_{t+1}$, and camera pose $\widetilde{h}'_{t+1}$. From $\widetilde{c}_t$, the kinematic policy $\pi_{\text{KIN}}^{\text{step}}$ computes the root angular velocity $\widetilde{w}_t$, linear velocity $\widetilde{v}_t$, and next frame joint rotation $\widetilde{j}_{t+1}^{\text{rot}}$. The next frame pose is computed through a finite integration module $T_{\text{finite}}$ with time difference $\delta t = 1/30 s$:

$$\widetilde{\omega}_t, \widetilde{v}_t, \widetilde{j}_{t+1}^{\text{rot}} = \pi_{\text{KIN}}^{\text{step}}(\widetilde{c}_t), \quad \widetilde{q}_{t+1} = T_{\text{finite}}(\widetilde{\omega}_t, \widetilde{v}_t, \widetilde{j}_{t+1}^{\text{rot}}, \widetilde{q}_t). \tag{1}$$

When trained without physics simulation, we auto-regressively apply the kinematic policy and use the computed $\widetilde{q}_{t+1}$ as the input for the next timestep. This procedure is outlined at Alg. 1. Since all mentioned calculations are end-to-end differentiable, we can directly optimize our $\pi_{\text{KIN}}^{\text{init}}$ and $\pi_{\text{KIN}}^{\text{step}}$

---

**Algorithm 1** Learning kinematic policy via supervised learning.

---

1: **Input:** Egocentric videos $I$ and paired ground truth motion dataset $\widehat{Q}$
2: **while** not converged **do**
3:     $M_{\text{SL}} \leftarrow \emptyset$                                                 ▷ initialize sampling memory
4:     **while** M not full **do**
5:         $I_{1:t} \leftarrow$ random sequence of images $I_{1:T}$ from the dataset $I$
6:         $\widetilde{q}_1, \phi_{1:T}, \widetilde{o}_1, \widetilde{h}_{1:T} = \pi^{\text{init}}_{\text{KIN}}(I_{1:T})$             ▷ compute scene context and initial pose
7:         **for** $i \leftarrow 1...T$ **do**
8:             $\widetilde{c}_t \leftarrow T_{\text{AC}}(\widetilde{q}_t, \phi_{t+1}, \widetilde{o}_{t+1}, \widetilde{h}_{t+1})$        ▷ compute agent-centric input features
9:             $\widetilde{q}_{t+1} \leftarrow T_{\text{finite}}(\pi^{\text{step}}_{\text{KIN}}(\widetilde{c}_t), \widetilde{q}_t)$
10:            store $(\widetilde{q}_t, \widehat{q}_t)$ into memory $M_{\text{SL}}$
11:         **end for**
12:     **end while**
13:     $\pi^{\text{step}}_{\text{KIN}}, \pi^{\text{init}}_{\text{KIN}} \leftarrow$ supervised learning update using data collected in $M_{\text{SL}}$ for 10 epoches.
14: **end while**

---

through supervised learning. Specifically, given ground truth $\widehat{q}_{1:T}$ and estimated $\widetilde{q}_{1:T}$ pose sequence, our loss is computed as the difference between the desired and ground truth values of the following quantities: agent root position ($\widehat{r}^{\text{pos}}_t$ vs $\widetilde{r}^{\text{pos}}_t$) and orientation ($\widehat{r}^{\text{rot}}_t$ vs $\widetilde{r}^{\text{rot}}_t$), agent-centric object position ($\widehat{o}'^{\text{pos}}_t$ vs $\widetilde{o}'^{\text{pos}}_t$) and orientation ($\widehat{o}'^{\text{rot}}_t$ vs $\widetilde{o}'^{\text{rot}}_t$), and agent joint orientation ($\widehat{j}^{\text{rot}}_t$ vs $\widetilde{j}^{\text{rot}}_t$) and position ($\widehat{j}^{\text{pos}}_t$ vs $\widetilde{j}^{\text{pos}}_t$, computed using forward kinematics):

$$\mathcal{L}_{\text{SL}} = \sum_{i=1}^{T} \|\widetilde{r}^{\text{rot}}_t \ominus \widehat{r}^{\text{rot}}_t\|^2 + \|\widetilde{r}^{\text{pos}}_t - \widehat{r}^{\text{pos}}_t\|^2 + \|\widetilde{o}'^{\text{rot}}_t \ominus \widehat{o}'^{\text{rot}}_t\|^2 + \|\widetilde{o}'^{\text{pos}}_t - \widehat{o}'^{\text{pos}}_t\|^2 + \|\widetilde{j}^{\text{rot}}_t \ominus \widehat{j}^{\text{rot}}_t\|^2 + \|\widetilde{j}^{\text{pos}}_t - \widehat{j}^{\text{pos}}_t\|^2. \quad (2)$$

### 3.3 Dynamics-Regulated Training

To tightly integrate our kinematic and dynamics models, we design a *dynamics-regulated training* procedure, where the kinematic policy learns from explicit physics simulation. In the procedure described in the previous section, the next-frame pose fed into the network is computed through finite integration and is not checked by physical laws: whether a real human can perform the computed pose is never verified. Intuitively, this amounts to mentally think about moving in a physical space *without actually moving*. Combining our UHC and our kinematic policy, we can leverage the prelearned motor skills from UHC and let the kinematic policy act directly in a simulated physical space to obtain feedback about physical plausibility. The procedure for dynamics-regulated training is outlined in Alg. 2. In each episode, we use $\pi^{\text{init}}_{\text{KIN}}$ and $\pi^{\text{step}}_{\text{KIN}}$ as in Alg. 1, with the key distinction being: at the next timestep $t + 1$, the input to the kinematic policy is the result of UHC and physics simulation $q_{t+1}$ instead of $\widetilde{q}_{t+1}$. $q_{t+1}$ explicitly verify that the $\widetilde{q}_{t+1}$ produced by the kinematic policy can be successfully followed by a motion controller. Using $q_{t+1}$ also informs our $\pi^{\text{step}}_{\text{KIN}}$ of the current humanoid state and encourages the policy to adjust its predictions to improve humanoid stability.

**Dynamics-regulated optimization.** Since the physics simulation is not differentiable, we cannot directly optimize the simulated pose $q_t$; however, we can optimize $q_t$ through reinforcement learning and $\widetilde{q}_t$ through supervised learning. Since we know that $\widehat{q}_t$ is a *good guess* reference motion for UHC, we can directly optimize $\widetilde{q}_t$ via supervised learning as done in Sec. 3.2 using the loss defined in Eq. 2. Since the data samples are collected through physics simulation, the input $q_t$ is physically-plausible and more diverse than those collected purely through auto-regressively applying $\pi^{\text{step}}_{\text{KIN}}$ in Alg. 1. This way, our dynamics-regulated training procedure performs a powerful data augmentation step, exposing $\pi^{\text{step}}_{\text{KIN}}$ with diverse states collected from simulation.

However, MoCap pose $\widehat{q}_t$ is imperfect and can contain physical violations itself (foot-skating, penetration *etc.*), so asking the policy $\pi^{\text{step}}_{\text{KIN}}$ to produce $\widehat{q}_t$ as reference motion *regardless of the current humanoid state* can lead to instability and cause the humanoid to fall. The kinematic policy should adapt to the current simulation state and provide reference motion $\widetilde{q}_t$ that can lead to poses similar to $\widehat{q}_t$ yet still physically-plausible. Such behavior will not emerge through supervised learning and require *trial and error*. Thus, we optimize $\pi^{\text{step}}_{\text{KIN}}$ through reinforcement learning and reward maximization. We design our RL reward to have two components: motion imitation and dynamics self-supervision. The motion imitation reward encourages the policy to match the computed camera

---

**Algorithm 2** Learning kinematic policy via dynamics-regulated training.

---
1: **Input:** Pre-trained controller $\pi_{\text{UHC}}$, egocentric videos $I$, and paired ground truth motion dataset $\widehat{Q}$
2: Train $\pi_{\text{KIN}}^{\text{init}}, \pi_{\text{KIN}}^{\text{step}}$ using Alg. 1 for 20 epoches (optional).
3: **while** not converged **do**
4:     $M_{\text{dyna}} \leftarrow \emptyset$                                                     ▷ initialize sampling memory
5:     **while** $M_{\text{dyna}}$ not full **do**
6:         $I_{1:t} \leftarrow$ random sequence of images $I_{1:T}$
7:         $q_1 \leftarrow \widetilde{q}_1, \phi_{1:T}, \widetilde{o}_1, \widetilde{h}_{1:T} \leftarrow \pi_{\text{KIN}}^{\text{init}}(I_{1:T})$          ▷ compute scene context and initial pose
8:         $s_1 \leftarrow (q_1, \dot{q}_1)$                                     ▷ compute intial state for simulation
9:         **for** $i \leftarrow 1...T$ **do**
10:             $c_t \leftarrow T_{\text{AC}}(q_t, \phi_{t+1}, \widetilde{o}_{t+1}, \widetilde{h}_{t+1})$    ▷ compute agent-centric features using simulated pose $q_t$
11:             $\widetilde{q}_{t+1} \sim T_{\text{finite}}(\pi_{\text{KIN}}^{\text{step}}(c_t), q_t)$                  ▷ sample from $\pi_{\text{KIN}}^{\text{step}}$ as a guassian policy
12:             $s_t \leftarrow (q_t, \dot{q}_t)$
13:             $s_{t+1} \leftarrow P_{\text{physics}}(s_{t+1}|s_t, a_t), a_t \leftarrow \pi_{\text{UHC}}(a_t|s_t, \widetilde{q}_{t+1})$     ▷ phyics simulation using $\pi_{\text{UHC}}$
14:             $q_{t+1} \leftarrow s_{t+1}, r_t^{\text{KIN}} \leftarrow$ reward from Eq. 3    ▷ extract reward and $q_{t+1}$ from simulation
15:             store $(s_t, a_t, r_t, s_{t+1}, \widehat{q}_t, \widetilde{q}_{t+1})$ into memory $M_{\text{dyna}}$
16:         **end for**
17:     **end while**
18:     $\pi_{\text{KIN}}^{\text{step}} \leftarrow$ Reinforcement learning updates using experiences collected in $M_{\text{dyna}}$ for 10 epoches.
19:     $\pi_{\text{KIN}}^{\text{init}}, \pi_{\text{KIN}}^{\text{step}} \leftarrow$ Supervised learning update using experiences collected in $M_{\text{dyna}}$ for 10 epoches.
20: **end while**

---

trajectory $\widetilde{h}_t$ and MoCap pose $\widehat{q}_t$, and serves as a regularization on motion imitation quality. The dynamics self-supervision reward is based on the insight that the disagreement between $\widetilde{q}_t$ and $q_t$ contains important information about the quality and physical plausibility of $\widetilde{q}_t$: the better $\widetilde{q}_t$ is, the easier it should be for UHC to mimic it. Formally, we define the reward for $\pi_{\text{KIN}}^{\text{step}}$ as:

$$
\begin{aligned}
r_t = {}& w_{\text{hp}} e^{-45.0(\|h^{\text{pos}}_t - \widetilde{h}^{\text{pos}}_t\|^2)} + w_{\text{hq}} e^{-45.0(\|h^{\text{rot}}_t \ominus \widetilde{h}^{\text{rot}}_t\|^2)} + w_{\text{jv}}^{\text{gt}} e^{-0.005(\|\dot{j}^{\text{rot}}_t \ominus \widehat{\dot{j}^{\text{rot}}_t}\|^2)} + \\
& w_{\text{jr}}^{\text{gt}} e^{-50.0(\|j^{\text{rot}}_t \ominus \widehat{j}^{\text{rot}}_t\|^2)} + w_{\text{jr}}^{\text{dyna}} e^{-50.0(\|j^{\text{rot}}_t \ominus \widetilde{j}^{\text{rot}}_t\|^2)} + w_{\text{jp}}^{\text{dyna}} e^{-50.0(\|j^{\text{pos}}_t - \widetilde{j}^{\text{pos}}_t\|^2)},
\end{aligned}
\tag{3}
$$

$w_{\text{hp}}, w_{\text{hq}}$ are weights for matching the extracted camera position $\widetilde{h}^{\text{pos}}_t$ and orientation $\widetilde{h}^{\text{rot}}_t$; $w_{jr}^{\text{gt}}, w_{jv}^{\text{gt}}$ are for matching ground truth joint angles $\widehat{j}^{\text{rot}}$ and angular velocities $\widehat{\dot{j}^{\text{rot}}_t}$. $w_{\text{jr}}^{\text{dyna}}, w_{\text{jp}}^{\text{dyna}}$ are weights for the dynamics self-supervision rewards, encouraging the policy to match the target kinematic joint angles $\widetilde{j}^{\text{rot}}_t$ and positions $\widetilde{j}^{\text{pos}}_t$ to the simulated joint angles $j^{\text{rot}}_t$ and positions $j^{\text{pos}}_t$. As demonstrated in Sec. 4.2, the RL loss is particularly helpful in adapting to challenging real-world sequences, which requires the model to adjust to domain shifts and unseen motion.

**Test-time.** At the test time, we follow the same procedure outlined in Alg 2 and Fig.2 to roll out our policy to obtain simulated pose $q_{1:T}$ given a sequence of images $I_{1:T}$. The difference being instead of sampling from $\pi_{\text{KIN}}^{\text{step}}(c_t)$ as a Guassian policy, we use the mean action directly.

## 4   Experiments

**Datasets.** As no public dataset contains synchronized ground-truth full-body pose, object pose, and egocentric videos with human-object interactions, we record two egocentric datasets: one inside a MoCap studio, another in the real-world. The MoCap dataset contains 266 sequences (148k frames) of paired egocentric videos and annotated poses. It features one of the five actions: sitting down on a chair, avoiding obstacles, stepping on a box, pushing a box, and generic locomotion [55] (walking, running, crouching) recorded using a head-mounted GoPro. Each action has around 50 sequences with different starting position and facing, gait, speed *etc*. We use an 80–20 train test data split on this MoCap dataset. The real-world dataset is only for testing purpose and contains 183 sequences (50k frames) of an additional subject performing similar actions in an everyday setting wearing a head-mounted iPhone. For both datasets, we use different objects and varies the object 6DoF pose for each capture take. Additional details (diversity, setup *etc*.) can be found in Appendix D.

**Evaluation metrics.** We use both pose-based and physics-based metrics for evaluation. To evaluate the 3D global pose accuracy, we report the root pose error ($E_{\text{root}}$) and root-relative mean per joint position error [23] ($E_{\text{mpjpe}}$). When ground-truth root/pose information is unavailable (for real-world

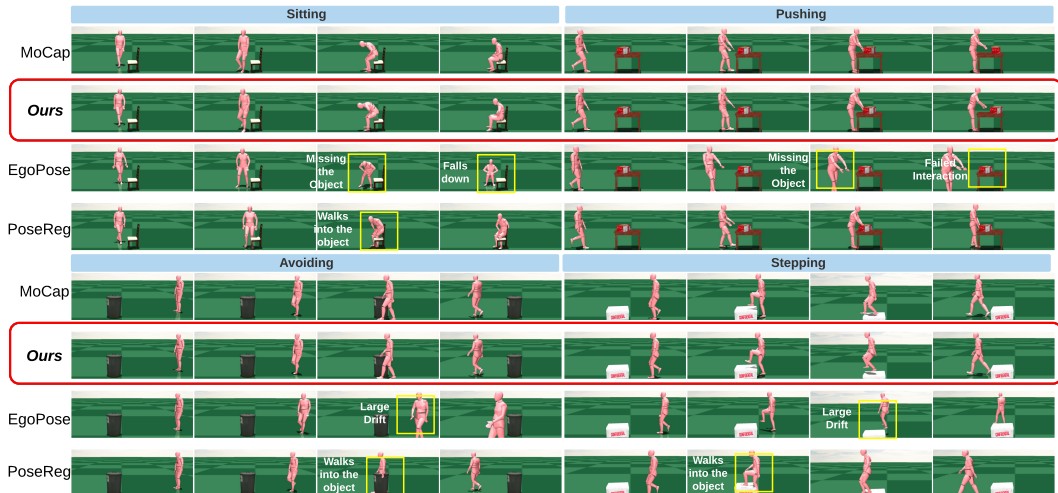

Figure 3: Results of egocentric pose and human-object interaction estimation from the MoCap datset.

dataset), we substitute $E_{root}$ with $E_{cam}$ to report camera pose tracking error. We also employ four physics based pose metrics: acceleration error ($E_{acc}$), foot skating ( FS), penetration (PT), and interaction success rate ($S_{inter}$). $E_{acc}$ (mm/frame$^2$) compares the ground truth and estimated average joint acceleration; FS (mm) is defined the same as in Ling *et al.* [26]; PT (mm) measures the average penetration distance between our humanoid and the scene (ground floor and objects). Notice that our MoCap dataset has *an penetration of 7.182 mm and foot sliding of 2.035 mm* per frame, demonstrating that the MoCap data is imperfect and may not serve as the best target motion. $S_{inter}$ is defined as whether the objects of interest has been moved enough (pushing and avoiding) or if desired motion is completed (stepping and sitting). If the humanoid falls down at any point, $S_{inter} = 0$. For a full definition of our evaluation metrics, please refer to Appendix B.

**Baseline methods.** To show the effectiveness of our framework, we compare with the previous state-of-the-art egocentric pose estimation methods: (1) the best dynamics-based approach *EgoPose* [56] and (2) the best kinematics-based approach *PoseReg*, also proposed in [56]. We use the official implementation and augment their input with additional information ($\widetilde{o}_t$ and $\widetilde{h}_t$) for a fair comparison. In addition, we incorporate the fail-safe mechanism [56] to reset the simulation when the humanoid loses balance to ensure the completion of each sequence (details in Appendix B).

**Implementation details.** We use the free physics simulator MuJoCo [47] and run the simulation at 450 Hz. Our learned policy is run every 15 timesteps and assumes all visual inputs are at 30 Hz. The humanoid follows the kinematic and mesh definition of the SMPL model and has 25 bones and 76 DoF. We train our method and baselines on the training split (202 sequences) of our MoCap dataset. The training process takes about 1 day on a RTX 2080-Ti with 35 CPU threads. After training and the initialization step, our network is causal and runs at 50 FPS on an Intel desktop CPU. For more implementation details and network architecture, please refer to Appendix B and C.

### 4.1 Results

**MoCap dataset results.** Table 1 shows the quantitative comparison of our method with the baselines. All results are averaged across five actions and all models have access to the same inputs. We observe that our method, trained either with supervised learning or dynamics-regulated, outperform the two state-of-the-art methods across all metrics. Not surprisingly, our purely kinematic model performs the best on pose-based metrics, while our dynamics-regulated trained policy excels at the physics-based metrics. Comparing the kinematics-only models we can see that our method has a much lower (79.4% error reduction) root and joint position error (62.1% error reduction) than PoseReg, which shows that our object-aware and autoregressive design of the kinematic model can better utilize the provided visual and scene context and avoid compounding errors. Comparing with the dynamics-based methods, we find that the humanoid controlled by EgoPose has a much larger root drift, often falls down to the ground, and has a much lower success rate in human-object interaction (48.4 % vs 96.9%). Upon visual inspection in Fig. 3, we can see that our kinematic policy can faithfully produce human-object interaction on almost every test sequence from our MoCap dataset, while PoseReg

Table 1: Quantitative results on pose and physics based metrics on the MoCap and real-world Dataset.

| | | | | | | | | |
|---|---|---|---|---|---|---|---|---|
| MoCap dataset | | | | | | | | |
| Method | Physics | $S_{inter}\uparrow$ | $E_{root}\downarrow$ | $E_{mpjpe}\downarrow$ | $E_{acc}\downarrow$ | FS $\downarrow$ | PT $\downarrow$ | |
| PoseReg | ✗ | - | 0.857 | 87.680 | 12.981 | 8.566 | 42.153 | |
| Kin_poly: supervised learning (ours) | ✗ | - | **0.176** | **33.149** | **6.257** | 5.579 | 10.076 | |
| EgoPose | ✓ | 48.4% | 1.957 | 139.312 | 9.933 | 2.566 | 7.102 | |
| Kin_poly: dynamics-regulated (ours) | ✓ | **96.9%** | 0.205 | 40.443 | 7.064 | **2.474** | **0.686** | |

| | | | | | | | | |
|---|---|---|---|---|---|---|---|---|
| Real-world dataset | | | | | | | | |
| Method | Physics | $S_{inter}\uparrow$ | $E_{cam}\downarrow$ | FS $\downarrow$ | PT $\downarrow$ | Per class success rate $S_{inter}\uparrow$ | | |
| PoseReg | ✗ | - | 1.260 | 6.181 | 50.414 | Sit | Push | Avoid | Step |
| Kin_poly: supervised learning (ours) | ✗ | - | 0.491 | 5.051 | 34.930 | | | | |
| EgoPose | ✓ | 9.3% | 1.896 | **2.700** | 1.922 | 7.93% | 6.81% | 4.87% | 0.2% |
| Kin_poly: dynamics-regulated (ours) | ✓ | **92.3%** | **0.476** | 2.742 | **1.229** | **98.4%** | **90.9%** | **100%** | **74.2%** |

and EgoPose often miss the object-of-interest (as can be reflected by the large root tracking error). Both of the dynamics-based methods has smaller acceleration error, foot skating, and penetration; some even smaller than MoCap (which has 2 mm FS and 7mm PT). Notice that our joint position error is relatively low compared to state-of-the-art third-person pose estimation methods [23, 24, 28] due to our strong assumption about known object of interest, its class, and potential human-object interactions, which constrains the ill-posed problem pose estimation from just front-facing cameras.

**Real-world dataset results.** The real-world dataset is far more challenging, having similar number of sequences (183 clips) as our training set (202 clips) and recorded using different equipment, environments, and motion patterns. Since no ground-truth 3D poses are available, we report our results on camera tracking and physics-based metrics. As shown in Table 1, our method outperforms the baseline methods by a large margin in almost all metrics: although EgoPose has less foot-skating (as it also utilizes a physics simulator), its human-object interaction success rate is extremely low. This can be also be reflected by the large camera trajectory error, indicating that the humanoid is drifting far away from the objects. The large drift can be attributed to the domain shift and challenging locomotion from the real-world dataset, causing EgoPose's humanoid controller to accumulate error and lose balance easily. On the other hand, our method is able to generalize and perform successful human-object interactions, benefiting from our pretrained UHC and kinematic policy's ability to adapt to new domains and motion. Table 1 also shows a success rate breakdown by action. Here we can see that "stepping on a box" is the most challenging action as it requires the humanoid lifting its feet at a precise moment and pushing itself up. Note that our UHC has never been trained on any stepping or human-object interaction actions (as AMASS has no annotated object pose) but is able to perform these action. As motion is best seen in videos, we refer readers to our supplementary video.

## 4.2 Ablation Study

To evaluate the importance of our components, we train our kinematic policy under different configurations and study its effects on the *real-world dataset*, which is much harder than the MoCap dataset. The results are summarized in Table 2. Row 1 (R1) corresponds to training the kinematic policy only with Alg. 1 only and use UHC to mimic the target kinematic motion as a post-processing step. Row 2 (R2) are the results

Table 2: Ablation study of different components of our framework.

| Component | | | | Metric | | | |
|---|---|---|---|---|---|---|---|
| SL | Dyna_reg | RL | VIO | $S_{inter}\uparrow$ | $E_{cam}\downarrow$ | FS $\downarrow$ | PT $\downarrow$ |
| ✓ | ✗ | ✗ | ✓ | 73.2% | 0.611 | 4.234 | 1.986 |
| ✓ | ✓ | ✗ | ✓ | 80.9% | 0.566 | 3.667 | 4.490 |
| ✓ | ✓ | ✓ | ✗ | 54.1% | 1.129 | 7.070 | 5.346 |
| ✓ | ✓ | ✓ | ✓ | **92.3%** | **0.476** | **2.742** | **1.229** |

of using dynamics-regulated training but only performs the supervised learning part. R3 show a variant trained without the estimated camera pose from VIO. Comparing R1 and R2, the lower interaction success rate (73.2% vs 80.9%) indicates that exposing the kinematic policy to states from the physics simulation serves as a powerful data augmentation step and leads to a model more robust to real-world scenarios. R2 and R4 show the benefit of the RL loss in dynamics-regulated training: allowing the kinematic policy to deviate from the MoCap poses makes the model more adaptive and achieves higher success rate. R3 and R4 demonstrate the importance of *intelligently* incorporating extracted camera pose as input: visual features $\phi_t$ can be noisy and suffer from domain shifts, and using techniques such as SLAM and VIO to extract camera poses as an additional input modality

can largely reduce the root drift. Intuitively, the image features computed from optical flow and the camera pose extracted using VIO provides similar set of information, while VIO provides a cleaner information extraction process. Note that our kinematic policy *without using* extracted camera trajectory outperforms EgoPose that *uses camera pose* in both success rate and camera trajectory tracking. Upon visual inspection, the humanoid in R3 largely does not fall down (compared to in EgoPose) and mainly attributes the failure cases to drifting too far from the object.

## 5 Discussions

### 5.1 Failure Cases and Limitations

Although our method can produce realistic human pose and human-object interaction estimation from egocentric videos, we are still at the early stage of this challenging task. Our method performs well in the MoCap studio setting and generalizes to real-world settings, but is limited to a *predefined set* of interactions where we have data to learn from. Object class and pose information is computed by off-the-shelf methods such as Apple's ARkit [17], and is provided as a strong prior to our kinematic policy to infer pose. We also only factor in the 6DoF object pose in our state representation and discard all other object geometric information. The lower success rate on the real-world dataset also indicates that our method still suffers from covariate shifts and can become unstable when the shift becomes too extreme. Our Universal Humanoid Controller can imitate everyday motion with high accuracy, but can still fail at extreme motion. Due to the challenging nature of this task, in this work, we focus on developing a general framework to ground pose estimation with physics by merging the kinematics and dynamics aspects of human motion. To enable pose and human-object interaction estimation for arbitrary actions and objects, better scene understanding and kinematic motion planning techniques need to be developed.

### 5.2 Conclusion and Future Work

In this paper, we tackle, for the first time, estimating physically-plausible 3D poses from an egocentric video while the person is interacting with objects. We collect a motion capture dataset and real-world dataset to develop and evaluate our method, and extensive experiments have shown that our method outperforms all prior arts. We design a dynamics-regulated kinematic policy that can be directly trained and deployed inside a physics simulation, and we purpose a general-purpose humanoid controller that can be used in physics-based vision tasks easily. Through our real-world experiments, we show that it is possible to estimate 3D human poses and human-object interactions from just an egocentric view captured by consumer hardware (iPhone). In the future, we would like to support more action classes and further improve the robustness of our method by techniques such as using a learned motion prior. Applying our dynamics-regulated training procedure to other vision tasks such as visual navigation and third-person pose estimation can also be of interest.

**Acknowledgements:** This project was sponsored in part by IARPA (D17PC00340), and JST AIP Acceleration Research Grant (JPMJCR20U1).

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
