# OpenReview forum: "Dynamics-regulated kinematic policy for egocentric pose estimation"
_NeurIPS.cc/2021/Conference — NeurIPS 2021 Poster_

### Official Review · Reviewer_7PSv · 2021-07-16

**Rating:** 7
**Confidence:** 4

**Summary:**

This paper looks at the problem of estimating physically-plausible 3D human motion and environment interactions from an egocentric RGB video. It proposes to combine the advantages of kinematic and dynamics-based approaches: first, a physics-based dynamics policy (UHC) is trained to imitate motion capture data; then a kinematic policy is trained to predict a target pose for UHC based on RGB and object & camera pose inputs. Experiments show that the combination of dynamics and kinematics outperforms either individually by a large margin, and that the method can reasonably predict a small set of person-object interactions.

**Limitations And Societal Impact:**

Limitations are addressed in the supp document and video - perhaps some of this discussion could be moved to the main paper.

The "Broader societal impact" section discusses valid privacy concerns.

**Main Review:**

Strengths:
-
The problem of physically-plausible 3D human motion (from ego or even 3rd person) is challenging, and even more so when considering environment and object interactions (which hasn't even been done much for 3rd person). Though some assumptions are made, the proposed approach makes a great first step towards solving this problem, producing impressive qualitative video results.

The technical approach leverages multiple good ideas that make it a compelling contribution. First, combining dynamic and kinematic approaches in a way that they are aware of each other, and so the dynamics policy can regularize the kinematic one. And second the notion of first learning a low-level physical motion controller on a diverse set of motions, and then leveraging this with a higher-level model for a problem with less available good data. The standalone dynamics policy (UHC) could also be useful for other applications.

Experiments report an extensive set of evaluation metrics that measure the kinematic, physical, and interaction accuracy of the proposed method. Quantitative and qualitative improvement is clear compared to SOTA kinematic-only & dynamics-only baselines and an ablation that naively uses dynamics to post-process a kinematics prediction, showing that the combination is advantageous.

The method also achieves a high success rate when estimating (a limited set of 4) human-object interactions, showing it is not limited to non-interacting motions.

Weaknesses:
-
The method makes a few limiting assumptions in order to make the problem feasible - these are understandable considering the difficulty and novelty of the problem. First, it assumes that accurate detections and poses of interacting objects are feasible from the ego view, and the ego-pose pipeline overfits to a fairly small set of actions/objects (sit, push, avoid, step). In many of the supp video examples, the object is not visible from the camera view so these detections would be difficult (I'm not sure exactly how they were done in the real-world data?) - then the method needs to rely on given prior knowledge of the 3D environment or overfitting to the image/camera pose features during interactions. Second, the method relies on accurate estimation of camera pose, which although is "optional" significantly affects performance as in Table 2. The ability of the method to generalize so well to the real-world dataset (with very different image features), suggests it relies heavily on this input. It would be helpful to see qualitative examples of the ablation that doesn't use camera pose; I'm also curious to see results without any image inputs, only using camera pose, would it still do well?

A few small technical issues/questions. First, the PD controller gains must be manually specified - these could instead be learned as in SimPOE. Second, on L199, how is the kinematic policy expected to predict next frame pose/velocities (at $t+1$) without any inputs corresponding to the next frame? Is all necessary future information contained in the image features through optical flow? Why not use the object/camera pose and image features at $t+1$ as well? Otherwise, it seems to be blindly sampling one possible future rather than reconstructing the observations.

Lastly, it's clear that combining kinematics+dynamics is useful, but the experiments do not justify why the dynamics policy must be pre-trained. Could Alg 2 be used to train both the dynamics and kinematic policy jointly? I suspect this will result in much worse performance - I like the idea of pre-training dynamics on a much wider set of motions before using it for the specific ego-estimation task -  but this baseline would empirically justify the pre-trained general UHC.

Questions/Suggestions:
-
L211: why is object position supervised in Eqn (2)? How is object position deviating from the initial detections if no physics are used in the regular supervised setting?

L288 says the method is causal and runs at 50 FPS, but the initial pose estimation from $\pi_{KIN}^{init}$ uses all future timesteps. How is this done in a real-time setting? Also in Alg 2, L7: how is the initial velocity $\dot{q}_1$ calculated? Is this predicted along with the pose by the initialization model?

Will the mocap and real-world datasets be released publicly?

The mocap dataset contains three different subjects, but it's not clear how the proposed method can handle differing body shapes and proportions since they are not given as input. Is the method currently simply overfitting to the heights of these three people?

Names like "Kin_poly: supervised learning (ours)" in Table 1 are a bit cumbersome, maybe consider giving a short, catchy name to the method variations.

Typos:
- Alg 1, L7 and Alg 2, L8: what is $n$? Should this be $t \leftarrow 1 \dots T$?
- Alg 2 , L14: does $\tilde{q}_{t+1}$ also need to be stored to compute the reward in Eq (3)?
- L228 differentialbe -> differentiable
- L288 casual -> causal
- Table 1 real-world data: shouldn't kin_poly supervised have an X in the "Physics" column?
- L310: phsycis->physics

=====================================

After Author Response:
-

I would like to thank the authors for their detailed responses to my and other reviewers' concerns and questions. After reviewing these responses and other reviews, I have kept my initial rating of accept.

Adding the updated quantitative evaluation of the UHC on AMASS to the paper/supp (using a __held out__ split of AMASS rather than the training data) and including more qualitative results on difficult motions make this contribution stronger and will be useful to the community, especially with code released. Furthermore, the idea of using UHC as physically-plausible regularization for inverse problems like pose estimation will be useful beyond ego-centric pose tracking.

As mentioned by the authors, the small set of motions and interactions in the ego-centric pose estimation task contribute to the very low errors, so the good performance on such an ill-posed task must be, at least in part, due to overfitting in some sense. Also, the requirement of the initialization procedure to acquire accurate object and head-camera poses further limits the generalizability of the current approach. Telling from Table 2 in the paper and followup experiments provided in the response wrt camera pose, I would encourage authors not to call the camera pose __optional__ since it is clearly important to good performance and admittedly the "framework mainly infer pose from egocentric motion".

However, I feel these limitations are understandable considering the difficulty of the new problem, and the results do still support the claim that dynamics-regulated training with UHC is useful. Moreover, the current formulation of the problem could be essentially seen as motion infilling [1] with only head pose observed and added object interactions to account for. This in itself is an open problem [2] with applications to e.g. animation in VR. The authors might consider discussing this in the revised paper, as I think the ideas they introduce are more generally useful to problems involving inferring physically-plausible motion from limited observations - not just ego-pose estimation.

[1] Convolutional Autoencoders for Human Motion Infilling, Kaufmann et al., 3DV 2020.

[2] LoBSTr: Real-time Lower-body Pose Prediction from Sparse Upper-body Tracking Signals, Yang et al., Eurographics 2021.



**Time Spent Reviewing:**

6

---

> ### Author Response · Authors · 2021-08-10
> **Author Response to Reviewer 7PSv**
>
> Thank you so much for your positive and supportive comments. We are glad to see that you find our work "a great first step", our qualitative result "impressive", and our technical approach a "compelling contribution". To address your concerns and comments:
>
> ---
>
> **Assumptions to make the problem feasible**
>
> We agree that our current framework makes a few important assumptions to make the problem solvable. Since capturing large-scale paired egocentric view and ground truth human pose is challenging, we focus on a common and important subset of human motion to develop viable approaches to incorporate a physics simulator and simulated character control for egocentric pose estimation. It serves as a benchmark for developing our universal humanoid controller and kinematic policy, as well as the optimization procedure needed to train the hybrid approach.
>
> To detect the objects in the real-world dataset, we use an off-the-shelf object pose estimator to estimate the object's 6 degrees-of-freedom pose. At the beginning of each sequence, the actors bow down slightly to point the head-mounted camera to the object to let the object pose estimator register the object of interest in the global space. We cut out this calibration period since it is irrelevant to the pose estimation task and may introduce unwanted biases. For each simulation episode, we use the estimated object pose to initialize the object simulation. All subsequent object movement is then a result of the human-object interaction. We will add these clarifying details to the dataset details in Appendix D.
>
> ---
>
> **Camera pose**
>
> The camera pose is optional in the sense that it provides a subset of information from the input video frames. The camera pose itself is extracted from the video frames using an off-the-shelf Visual Inertial Odometry (VIO) method. Our model infers human movement based on the motion information encoded in the video, so using the estimated camera pose serves as a direct and less noisy way of providing this information. Without camera pose, our method **can still estimate plausible human motion**. However,  global position estimation accuracy suffers significantly. As can be shown in Table 2, without using VIO the $E_{\text{cam}}$ degrades the most, which indicates that the humanoid drifts away from the video trajectory. This is expected since the model is conducting both SLAM/VIO as well as pose estimation in this setting. Upon visual inspection, the low success rate mainly attributes to humanoid drifting, missing the objects-of-interest or tripping on them. Notice that we train the previous state-of-the-art method with the same camera and object input, showing that our model can utilize this additional information more effectively. We will add additional qualitative examples without using camera pose in the supplementary video.
>
> We train an additional model that does not use the image features directly, and report the results:
>
> |       Real-world Dataset | | | | |
> | ----------- | ----------- | ----------- | ----------- | ----------- |
> | Method | $\text{S}_{\text{inter}} \uparrow$ | $\text{E}_{\text{cam}}\downarrow$  | $\text{FS} \downarrow$ | $\text{PT} \downarrow$ |
> | Kin_poly: Camera pose + Image features | **92.3%**  | **0.476**  | 2.742 | 1.229 |
> | Kin_poly: Image features only  | 54.1%  | 1.129  | 7.070 | 5.346 |
> | Kin_poly: Camera pose only  | 90.2%  | 0.500  | **2.688** | **1.005** |
>
> We can see that without using image features directly, the model can achieve comparable pose estimation results. This is expected as our framework mainly infer pose from egocentric motion, which is provided in the form of camera pose. Notice that the object states and camera pose are **extracted from the image inputs** using image-based methods, so the framework is still utilizing image inputs. The image features extracted from optical flow do provide redundant information and could be removed from the current pipeline. In the future, we aim to better utilize the egocentric view through more advanced scene context modeling, so incorporating image features would be of interest. In addition, given the potential use cases of this task (AR-glasses, body cams, etc.), we feel like reliable camera pose estimation is a reasonable assumption. We will update our manuscript to reflect these observations.
>
> ---
>
> **PD Controller**
>
> Thanks for the suggestions! Indeed, the gains can be learned through the meta-PD controllers as in SimPOE, and we have incorporated that in the newest version of the controller (though not included in the current submission).
>
> ---
>
> **Future input**
>
> We do not factor in any future information since it keeps our per-step model causal and compatible for future real-time pose estimation applications (currently our per-step model runs at 50 FPS). Our autoregressive kinematic policy can be viewed as a one-step pose estimation method: given the next-frame optical flow, camera pose, object states, and the current humanoid pose & velocity, it predicts the most probable velocities and poses. There is a typo in the current manuscript, where the correct input to the per-step model should have been $\widetilde{\boldsymbol c_t} = \boldsymbol T_{\text{AC}}(\widetilde{\boldsymbol q_t}, \boldsymbol \phi_{t+1}, \widetilde{\boldsymbol o_{t+1}}, \widetilde{\boldsymbol h}_{t+1})$. Since our kinematic policy is an RNN-based policy, it is predicting the most possible pose based on the next-frame observations. Due to the multi-modal and unconstrained nature of this task, it is possible that more sophisticated models will be needed for modeling more complex human motion, and we are excited to keep working on this challenging task.
>
> ---
>
> **Joint training kinematics and dynamics**
>
> The UHC benefited tremendously from learning from a large amount of human motion. By learning from motion sequences ranging from walking to kickboxing, our controller is robust to diverse and unseen input reference motion sequences. It also generalizes to lower-quality and noisy motion estimates from a learned pose estimation model. Since our MoCap dataset is small compared to AMASS (266 vs 11299 sequences), a controller jointly learned from our MoCap dataset would perform much worse than one learned from AMASS.
>
> To test out this hypothesis, we started training a joint-learning model when we received this comment. At the moment of this author's response (7 days after), the model has not yet converged. This can be largely attributed to the fact that the not pre-trained controller would not be able to produce a reasonably simulated pose even if the target pose from the kinematic policy is of good quality. The joint training process thus becomes unstable. As a comparison, our kinematic policy learned using a pre-trained UHC converges in 1 day. We will further verify this initial observation, debug the implementation, and include our findings in our manuscript.
>
> ---
>
> **Questions & suggestions**
>
> > L211, Object position
>
> Here the object position $\widehat{\boldsymbol{o}}^{\prime \text{pos}}_{t}$ is the agent-centric object position
> (the estimated/desired object position in the agentic centric coordinate frame). It calculates how far the agent is from the desired agent position by comparing the relative object positions, so it is very similar to the human root position and orientation loss. In the training MoCap dataset, we capture the object position and orientation using MoCap markers.
>
> > L288 and Alg.2 L7,  initialization
>
> Sorry for the confusion; it should have written: "After training and the initialization step…". Our per-step model runs at 50 FPS, but given the current setting, we rely on the initialization step to compute the initial pose. In the future, it is conceivable that the humanoid can be initialized through a calibration period or using a mean humanoid pose and the initialization model will no longer be needed. The camera, object state, and image pose can be inputted in a streaming fashion. We will correct this in the updated manuscript.
>
> The initial velocity is also calculated by the initialization model, we will update this accordingly.
>
> > Will the mocap and real-world datasets be released publicly?
>
> All of our code, data, and models will be released publicly for future research purposes.
>
> > How can the proposed method handle differing body shapes and proportions since they are not given as input?
>
> Currently, we use the SMPL mean body shape to simplify the body shape of the actors,  and we manually adjust the starting height of the MoCap pose to make sure that each of the humanoid’s feet is touching the ground at the starting point of the episode (L84, Appendix C). This process can be viewed as a normalization step of the actor's height and limb proportion. Since our universal humanoid controller is also trained using this adjusted data from AMASS, it is able to compensate for the difference in height. Our kinematic policy is also able to adapt to differences in motion induced by varying actor height. Since the Mujoco physics simulator requires the simulated humanoid's parameters to be prespecified as an XML file and can not be dynamically adjusted, we are actively developing methods that can utilize a more diverse set of body shapes.
>
> > Name suggestions, typos, etc:
>
> We thank you for all the suggestions and corrections and will incorporate all into our updated manuscript.
>
> - Alg 1, L7 and Alg 2, L8: yes, it should be T
>
> - Alg 2, L14: yes, $\widetilde{\boldsymbol{q}}_{t+1}$ is used to compute the mismatch between the target motion and UHC's output, to encourage the kinematic policy to output target motion that can be better mimicked by UHC. It is part of the dynamics self-supervision reward (L251).
>
> - Table 1 real-world data: yes, it should have been x; it is a physics-based model.

---

> > ### Comment · Reviewer_7PSv · 2021-08-15
> > **Clarification on "Future input"**
> >
> > Thank you for your detailed response.
> >
> > Just to ensure I am not misunderstanding re: using "future" inputs to do the pose estimation, I wanted to clarify. In the usual pose estimation/tracking formulation, we want to estimate the pose $q_t$ at timestep $t$ given the current observations $o_t$ (e.g. image, object state, camera pose, etc.) and the pose at the previous step $q_{t-1}$. So if the model is $\pi$ then $q_{t} = \pi(o_t, q_{t-1})$, i.e. we reconstruct the pose based on the given observations at that timestep. This can run in real time and operates on streaming input observations since it only requires the observations at the current step that we're estimating.
> >
> > My understanding of the proposed method (as detailed in Sec 3.2 of the paper) is that $\pi_{\text{KIN}}$ does not use the "current" observations and operates _only on past observations_, i.e. using the simplified notation above it is $q_{t} = \pi_{\text{KIN}}(o_{t-1}, q_{t-1})$. Is this correct? If so, the proposed approach is indeed predicting the future at each step, as you describe, and then repeatedly correcting itself after it gets $o_t$ as input for the following step $t+1$?
> >
> > This formulation does not seem to cause issues practically considering the limited set of motions, but I'm wondering what the motivation is for this since the task is pose estimation and not future motion prediction? The usual formulation can still run in real time and would avoid the mentioned multi-modality issue for more complex motions.
> >
> > Thanks!

---

> > > ### Author Response · Authors · 2021-08-17
> > > **Response to questions on "future input"**
> > >
> > > Thank you for your clarification question -- we have misunderstood your original question, where the use of the term "future" led us to some misunderstandings. We are predicting the next frame pose based on next frame input: $q_{t+t}  = \pi_{\text{KIN}}(o_{t+1}, q_t)$ (or equivalently, $q_{t}  = \pi_{\text{KIN}}(o_{t}, q_{t-1})$). There is indeed a notation issue in the current manuscript: the correct input to the per-step model (L199, Sec 3.2, along with a few other corresponding places) should have been $\widetilde{\boldsymbol c_t} = \boldsymbol T_{\text{AC}}(\widetilde{\boldsymbol q_t}, \boldsymbol \phi_{t+1}, \widetilde{\boldsymbol o_{t+1}}, \widetilde{\boldsymbol h}_{t+1})$, similar to SimPOE [1].
> > >
> > > To answer your original question, we use the next frame observation to predict the next frame pose, the same as in the standard pose estimation/tracking problem. We are not predicting the future based on current observations. This is also what happens in implementation, where our per-step model $\pi_{\text{KIN}}$ outputs the "current" pose (timestep t) based on the "current" observation (timestep t).
> > >
> > > In this setup, the problem is still multimodal and underconstrained, where the same set of egocentric observations can be the result of different human poses. Future work may benefit from outputting multiple possible current poses based on the current observation and use physics-based simulation to reason about possible human motion.
> > >
> > > We have updated our previous response and the manuscript to reflect these changes and clarifications. Thank you so much again for your questions and quick response! Please let us know if there is anything else we can clarify.
> > >
> > > [1] Ye Yuan, Shih-En Wei, Tomas Simon, Kris Kitani, and Jason Saragih, “Simpoe: Simulated character control for 3d human pose estimation”, CVPR, 2021

---

### Official Review · Reviewer_c7KC · 2021-07-17

**Rating:** 6
**Confidence:** 5

**Summary:**

This paper presents a method for estimating the egocentric 3D pose of a person moving in a scene. The main idea is to build a dynamics-driven human motion controller which is provided target poses to mimic by a kinematics estimator and information about scene objects. Experiments show that improved performance compared to previous approaches.

**Ethical Concerns:**

No.

**Limitations And Societal Impact:**

Limitations are discussed in the supplementary, but I would have liked to see this discussion in the main paper. The paper discusses broader discusses.

**Main Review:**

I would like to start with several things that I liked about the paper.

- The problem being solved is important and useful. Combining kinematics and dynamics is of increasing interest in the human pose estimation community. This paper takes it further by operating on egocentric data.
- The paper is well written, motivated and the experiments are comprehensive.
- The problem formulation is interesting and novel.

In the following, I focus on questions, comments, and suggestions for improvement.

- The introduction makes says (lines 33) that dynamics-based methods use physics simulators, but this is not always true. Many methods model dynamics explicitly and may not use a dynamics simulator.

- The introduction does not discuss the objects in much detail. It would have been nice to do that as this is a part of the proposed method.

- In the method section, I was lacking motivation for why certain components were used. For instance, what are the differences between the proposed controller vs. other methods that try to achieve the same thing as MotionVAE? The proposed approach appears fine but lacks insight and is more complex than other methods.

- In section 3.2, similarly for the kinematics model: Why not use a simple neural network to predict poses autoregressively?

- Line 183: How are object states encoded? How many objects are supported and does this generalize to unseen objects?

- For experiments, this might be a useful dataset to work on. The paper states no dataset fits but perhaps this could?

https://ps.is.tuebingen.mpg.de/publications/grab-2020

Overall, the paper presents some interesting ideas which I would like to see published. I encourage the authors to address the above points in their rebuttal.

**Time Spent Reviewing:**

2

---

> ### Author Response · Authors · 2021-08-10
> **Author Response to Reviewer c7KC**
>
> Thank you so much for your positive and helpful comments. We are happy to see that you find our task "important, useful, and novel"; our hybrid approach combining kinematics and dynamics "of interest to the community", and our paper "interesting and would like to see published". To answer your concerns and comments:
>
> ---
>
> **Use of a physics simulator**
>
> We agree that not all dynamics-model use physics simulators and there are many great works that directly use dynamics [1, 2, 3]. We will update the manuscript to reflect this observation. In this specific context (L31) we are referring to dynamics-based methods in egocentric pose estimation, which mostly utilize a physics simulator.
>
> ---
>
> **Discussion about objects**
>
> We will add more discussions about how we acquire and use the object and object states. Factoring in object and scene state is imperative to egocentric pose estimation since limited information can be extracted from an egocentric view alone. Since we use an off-the-shelf object pose estimator for detecting and extracting object states, we have focused our attention on the kinematics and dynamics part of the method during the introduction. While factoring object states is important to the performance of the proposed method, our main contribution lies in the universal humanoid controller, kinematic policy, and their symbiosis.
>
> ---
>
> **Proposed controller vs motionVAE**
>
> MotionVAEs [4] are purely kinematics-based methods that do not utilize a physics simulation and do not support human-object interaction. They aim to animate a character based on sparse control inputs. Compared to motionVAEs, our universal humanoid controller (UHC) aims to control a humanoid to imitate human motion inside a physics simulation. Through the use of a physics simulator, our UHC imposes a stricter constraint on the possible human motion: physically incorrect motions such as foot skating and penetration will cause the humanoid to fall down. Using a physics simulator also leads to realistic human-object interactions and the simulated character can react to external force naturally. We will add further discussion comparing our method to other controllers such as MotionVAEs.
>
> ---
>
> **Proposed kinematic model vs simple NN**
>
> On the kinematics side, our kinematic policy is an autoregressive pose estimator grounded with visual input. Abstractly, it is an autoregressive network that predicts pose at a per-frame level. The added complexity comes from the fact that we estimate pose in the global space and need to output the global position and orientation of the humanoid. In comparison, most third-person 3D pose estimation methods are root-relative and estimate human pose in the camera space. Our kinematic policy also factors in the visual context, object pose, and camera pose to estimate the most possible human pose based on egocentric observations. In order to generalize to unseen scene context and human-object spatial configurations, we need to transform all input to an agent-centric coordinate frame. In addition, we design our kinematic controller and UHC such that they share a significant common structure in their input and output space. Thus, we can train our kinematic policy with both reinforcement learning and supervised learning.
>
> ---
>
> **How are object states encoded?**
>
> We use an off-the-shelf object detector and pose estimator to extract the object states (L190, Sec. 3.2). The object states are represented by their class (one-hot vector) and 6 degrees-of-freedom (DOF) pose (global position and orientation). We then use an agent-centric transform $T_{\text{AC}}$ to convert its global pose into agent-relative pose. Currently, our framework can simulate an unbounded number of objects using the physics simulator, while the kinematic model factors in only the main object-of-interest (e.g. in the pushing action, we simulate both the table and the box, while the kinematic policy factors in the box's pose and class). Our framework can be easily expanded to factor in more objects-of-interest and our current use case only requires one. Since we only utilize the object class and 6DOF pose, our framework naturally generalizes to unseen objects of similar dimensions. Between the real-world dataset and MoCap dataset, we used different instances of chairs, boxes, obstacles, etc. We will add further details about our object state encoding to our manuscript.
>
> ---
>
> **The Grab dataset**
>
> Thank you so much for calling this dataset to our attention. This dataset is indeed very useful for future related projects in this area. Namely, we are very interested in simulating detailed hand motion and human-object interactions and are working toward this direction. For the task discussed in this submission, the **paired first-person view** would be necessary for the dataset to be used. One possibility is to generate synthetic image observation from the captured motion. Our current model also infers human motion mainly from head movement, so an updated method (maybe utilizing a different view or additional sensors) would be needed to estimate the motion present in the Grab dataset.
>
> ---
>
> **References**
>
> [1] Soshi Shimada, Vladislav Golyanik, Weipeng Xu, and C. Theobalt. Physcap: Physically plausible monocular 3d motion capture in real time. ACM Trans. Graph., 39:235:1–235:16, 2020.
>
> [2] Davis Rempe, L. Guibas, Aaron Hertzmann, Bryan C. Russell, R. Villegas, and Jimei Yang. Contact and human dynamics from monocular video. In SCA, 2020.
>
> [3] M. Vondrak, L. Sigal, J. Hodgins, and O. C. Jenkins. Video-based 3d motion capture through biped control. ACM Transactions on Graphics (TOG), 31:1 – 12, 2012.
>
> [4] Ling, Hung Yu, Fabio Zinno, George Cheng, and Michiel Van De Panne. 2020. “Character Controllers Using Motion VAEs.” ACM Transactions on Graphics 39 (4): 12.

---

### Official Review · Reviewer_cAMH · 2021-07-17

**Rating:** 7
**Confidence:** 4

**Summary:**

In this paper, the authors presents a new method for egocentric physically-plausible 3d human motion estimation. Specially, they designed a general-purpose humanoid controller (UHC) and a dynamics-regulated kinematic policy that can be directly trained and deployed inside a physics simulation. The proposed universal humanoid controller is trained on large-scale Mocap datasets and is for general-purpose. In the experiments, the authors evaluate their proposed method on both Mocap dataset and their collected real dataset. The authors show good performance improvements compared with PoseReg and EgoPose, as well as good imitation success rate compared with DeepMimic on Human3.6M.

**Limitations And Societal Impact:**

Overall this paper is well prepared. I found a few limitations in current method.
- Most of the testing data are quite simple and generally belong to walking/sitting poses. From the results reported in supplementary video and material, it seems the proposed methods cannot handle challenging poses well. It would be interesting to see how much the proposed method could work on more challenging poses. The authors could further expand their pose categories to more non-upright poses (e.g., working out, yoga, lying down) and this could probably give us more insights about the boundaries of the proposed method.
- I found the human-object interaction part is a little bit confusing. The objects are not reconstructed and tracked from the input video and we could see sometimes the object state in real-world video does not match it in simulation. What is the assumption here? If we wanna model real-world object interaction, how would you further leverage the current approach?
- Real-world testing data and experiments are still a little bit limited. The evaluation on Human3.6M is quite interesting. It would be interesting to dive deeper and analyze more about the failure cases and how the proposed policy could potentially improve to handle such failure cases.
- From the results I noticed the walking looks a little bit robotic and not quite like natural walking gait. Could you explain what causes this and how would you improve to make the movement more natural?

**Main Review:**

Overall, this paper is well written and easy to follow. The authors presents an interesting approach for egocentric physically-plausible 3d human motion estimation.
+ This paper is a re-submission of their AAAI work. The authors addressed the major concerns from their previous submission in method novelty compared with [Ye et al, ICCV'19], [Jungdam et al, TOG'20], performance significance and real-world dataset scale.
+ The proposed general-purpose humanoid controller and the designed dynamics-regulated kinematic policy is quite interesting.
+ The authors conduct quite some experiments to validate the method effectiveness and elaborate the implementation and experiment details well.
+ Code is attached in supplementary.

**Time Spent Reviewing:**

3

---

> ### Author Response · Authors · 2021-08-10
> **Author Response to Reviewer cAMH**
>
> We thank the reviewer for the positive and constructive comments. We are glad that you find our paper "well written and easy to follow", our controller and kinematic policy "quite interesting", and our experiments "effective". To answer your concerns and questions:
>
> ---
>
> **Limitation on testing data**
>
> Currently, we focus on modeling a subset of human actions (namely, sitting, pushing, stepping, avoiding, and locomotion) as a starting point to the challenging task of egocentric pose estimation that involves human-object interactions. Modeling the kinematics and dynamics of non-periodic motion such as sitting [1], turning, and stepping on a box involve dynamic sequences of motion that can easily lead to the humanoid tripping and falling down. This problem is exacerbated by the more diverse motion sequences recorded in the real-world dataset, where the speed, gait, and object configuration can be never observed in the MoCap training data. From the reported quantitative and qualitative results we can see that our method can generalize to unseen data well and extrapolate realistic human motion based on the laws of physics. While large variations in the egocentric videos' speed and gait can still be challenging, our model can generalize well to reasonable degrees of free-formed motion (as can be seen in timestamps 03:40, 01:20, and 0:48 from the supplementary video). We agree that the assumption of known object and human-object interaction is made to render the task solvable, but hope that our first attempt at this challenging task provides some solutions to estimate physically valid human motion and human-object interactions from egocentric videos.
>
> We are excited about and are working on expanding our method to non-upright poses such as working out, yoga, and lying down. Poses that involve significant and complex arm movement are especially important yet challenging as arms and legs are not always visible in an egocentric view. In those cases, it is possible that additional prior such as scene geometry or the camera wearer's intention would be necessary to achieve accurate pose estimation
>
>
> ---
>
> **Human-object interaction**
>
> We use an off-the-shelf object detector and pose estimator to extract the object 6 degrees-of-freedom pose *at the beginning* of each sequence's recording session (L190, Sec. 3.2). To keep the object-of-interest in the field of view, the actors bow down to point the head-mounted camera to the objects. We cut out this calibration period to avoid introducing bias. We use the estimated object pose to **initialize** the objects inside the simulation and subsequent object movements are all resulting from the simulated character. The difference between the object states can be attributed to the mismatch between physics simulation and the real-world, as well as discrepancies in the actor's actions. We will update our manuscript to further illustrate our data capturing procedure.
>
> We believe that there is great potential in using simulation for object characteristics and human-object interaction estimation: the discrepancy in simulated results and the real-world observation can serve as important signals for system identification and state estimation. By relating and analyzing the difference between simulation and real-world visual observation, we can update simulation parameters and humanoid control signals to more accurately reproduce observed human-object interactions. The proposed UHC, kinematic policy, and dynamics-regulated training can all be useful for this purpose.
>
> ---
>
> **Real-world data**
>
> We agree that more data collection and a more detailed analysis of the current method are crucial to advancing this challenging task. Currently, our real-world dataset is comparable in size with our whole training set (183 vs 202 sequences), while the real-world dataset has more diverse trajectories. It serves as a benchmark for testing the generalization of our method learned only using MoCap data and showcases the effectiveness of our framework to learn a policy capable of handling diverse human movement patterns. For the proposed method, failure cases exist mostly in the stepping action (as can be seen in the low success rate reported in Table 1) where the humanoid can easily trip on the box and fall down. Abrupt turning and leaning forward can also lead to instability.  We will expand upon the current failure case analysis in the manuscript.
>
> ---
>
> **Walking gait**
>
> Thank you for your close observation! There can be a couple of causes to this issue. One reason can be that the head-mount for the egocentric camera is shaking due to the actor's movement (this is especially apparent in the real-world dataset). The model tries to match such slight movement and walks unnaturally. It is also conceivable that the actor himself/herself is walking a bit unnaturally due to the added head-mounted camera. Another cause can be that our method is currently optimized through motion imitation reward and L2 loss, and does not take into account the naturalness of the motion. To improve the naturalness we could 1) collect more egocentric data and improve capture equipment, 2) incorporate methods such as AMP [2] to penalize unnatural kinematic motion.
>
>
> ---
>
> **References**
>
> [1] Chao, Yu-Wei, Jimei Yang, Weifeng Chen and Jia Deng. “Learning to Sit: Synthesizing Human-Chair Interactions via Hierarchical Control.” AAAI (2021).
>
> [2] Xue Bin Peng, Ze Ma, Pieter Abbeel, Sergey Levine, and Angjoo Kanazawa. 2021. AMP: adversarial motion priors for stylized physics-based character control. ACM Trans. Graph. 40, 4, Article 144 (August 2021), 20 pages.

---

### Official Review · Reviewer_ZmMv · 2021-07-17

**Rating:** 6
**Confidence:** 5

**Summary:**

This paper proposes a method for pose estimation from first-person camera view. The work builds on a universal controller that learns to imitate a wide variety of behaviors. A kinematic policy is then either trained fully via supervised learning or in a hybrid manner, using a combination of supervised learning and reinforcement learning. The reinforcement learning optimization consists of a motion imitation and a physics-based objective. Specifically, the universal controller’s output is utilized to regularize the kinematic policy to become more physically plausible. The main contributions are a universal controller and the dynamics-regulated training procedure for egocentric pose estimation in human-object interaction scenarios.

**Limitations And Societal Impact:**

Yes, the limitations and societal impact is discussed properly.

**Main Review:**

### Strengths ###

The paper is written in a clear manner and simple to follow. It introduces the new setting of egocentric pose estimation with objects in the scene. The main idea of using a physics-based policy directly to guide the training of the kinematics-based model is novel. In prior works [1][2][3], the policy was only used to correct the kinematics model.

### Weakness ###

Overview of the main concerns, which are detailed in the paragraphs below:
1. Improper evaluation of the universal controller.
2. Missing insights into the data and the strong performance of the pose estimation, which perform better than state-of-the-art pose estimation from third-person view on established benchmarks, despite the much harder task.
3. Generalization of the learned tasks is not properly evaluated.


**1. Universal controller**

First of all, learning robust universal controllers is a challenging task and is known to suffer from instabilities, especially if it has to replicate unseen motions. There is a branch of research [4][5][6] with the sole focus on building controllers that can scale to larger datasets, but have only achieved learning on a subset of AMASS (CMU ~ 2k motion sequences). The proposed method builds on a universal controller that is trained on AMASS (~11k sequences). Hence, the claim of a universal controller that can scale to an order of magnitude larger motion database than state-of-the-art needs to be properly evaluated on its own, since it would mark a substantial improvement in the direction of imitation learning from motion capture. As stated by the authors, the policy is able to execute a wide variety of motion “ranging from dancing to kickboxing” (cf. L.40) and therefore needs to be empirically evaluated on the full AMASS dataset to substantiate this claim. The current controller is only tested on the relatively simple motions contained in H36M (walking, standing, etc.). Furthermore, it is important to see how well it adapts to noisy estimates, because at the beginning of training the dynamic-regulated model, the kinematics policy will likely produce random residuals and hence noisy target poses.

**2. Missing insights into data and performance**

The universal controller is then utilized in the downstream task of egocentric pose estimation. Since the task is ill-posed, estimating pose from egocentric video without top-down view is extremely difficult. The results, namely the mean per joint position error, which is very low (~30-40mm), may indicate that there is very little variation between different motion sequences and overfitting to the training data (which is likely similar to the test data). For instance, since there is no way to tell how the occluded arms are moving, such a small error is only possible if the deviation of motion between the sequences in the data is very small. The authors should provide: 1) statistical data on their dataset to be able to better assess their quantitative results (e.g., data on per joint trajectories, pose diversity against available 3D datasets), 2) a proper discussion of the very small reported errors on the test data and 2) the per joint error to be able to get better insights into the performance of the model.
This seems even more evident when looking at the supplementary video, where it appears that the agent is always moving in the same way (movement of arms, gait, speed, etc.). Since the authors already provide the statistical data of the speed in the datasets, it would be good to see whether the agent actually learns to adapt to different speeds or just overfits to one single motion.

**3. Generalization**

As illustrated in the details of the dataset (Appendix D), the interaction objects are always positioned in the same location for both datasets, and hence it would be important to conduct experiments on generalization. For instance, how would the method fare if the objects’ locations were slightly different? Otherwise, the learned policy is likely to overfit to single object instances and not be useful for downstream applications. This should ideally be tested and at least be discussed in the paper. Furthermore, although the trajectory analysis provided in Figure 5 indicates a slight variation in the facing direction towards the object, it seems that the agent is mostly facing the interaction objects and needs to walk straight to reach it (cf. supplementary video). It would be interesting to see whether the agent can act in the scene without being right in front of the object of interest and facing towards it.


### Other Comments and Technical Questions ###
- The training procedure of the universal controller proposed for sampling sequences likely impose quite a large computational overhead since it has to run all frames of AMASS (4000k) through the value function. How often is this distribution recomputed? Moreover, is there a mistake in the notation of the initialization states, or what is the reason that the target state is the same as the input state and not the next state?

- Did the authors run experiments without the redundancy of information in the state space of the universal controller (e.g. to have joint angles in axis angle and quaternion representation or the difference between joint position and target joint position in world and agent-centric coordinates) or what was the incentive behind overloading the state space?

- Do the authors employ any technique, such as early stopping, when training the universal controller and the dynamics-regulated kinematic policy to avoid learning from failure cases?

- The notation in Appendix C Policy Network architecture seems to be inconsistent. The quaternion difference $\ominus$ and the minus seem to be used for angle-axis difference and vice-versa, at least according to the dimensions provided.

- Will the authors release the dataset in case of publication to foster further research?

- A potential missed citation is [7]. The authors use a differentiable physics model to correct their kinematics model for pose reconstruction.

[1] Ye Yuan, Shih-En Wei, Tomas Simon, Kris Kitani, and Jason Saragih, “Simpoe: Simulated character control for 3d human pose estimation”, CVPR, 2021

[2] Soshi Shimada, Vladislav Golyanik, Weipeng Xu, and Christian Theobalt. 2020, “PhysCap: physically plausible monocular 3D motion capture in real time”, ACM Transactions on Grap., 2020.

[3] Davis Rempe, Leonidas J Guibas, Aaron Hertzmann, Bryan Russell, Ruben Villegas, and Jimei Yang, “Contact and human dynamics from monocular video”, ECCV 2020

[4] Jungdam Won, Deepak Gopinath, and Jessica Hodgins, “A scalable approach to control diverse behaviors for physically simulated characters”, ACM Trans. Graph., 2020.

[5] ] Tingwu Wang, Yunrong Guo, Maria Shugrina, and Sanja Fidler, “Unicon: Universal neural controller for physics-based character motion”, arxiv, abs/2011.15119, 2020.

[6] Josh Merel, Leonard Hasenclever, Alexandre Galashov, Arun Ahuja, Vu Pham, Greg Wayne, Yee Whye Teh, and Nicolas Heess, “Neural probabilistic motor primitives for humanoid control”, ICLR, 2019

[7] Soshi Shimada, Vladislav Golyanik,  Weipeng Xu, Patrick Pérez, and Christian Theobalt, “"Neural PhysCap" Neural Monocular 3D Human Motion Capture with Physical Awareness”, ACM Trans. Graph., 2021.

### Post rebuttal
I appreciate the author's extensive answer to my concerns and questions. In the light of how the concerns were addressed, I'm willing to raise my score to 6. What I would like to see in the final version of the paper in case of acceptance is 1) a thorough discussion of the limitations (which is missing in the current main part of the manuscript) 2) the requested evaluations of the universal controller and the dataset statistics. 3) Clarification of how the pose metrics were obtained with respect to failing sequences. It seems that such good numbers for the pose can only be achieved if the sequences are ended for episodes deemed unsuccessful.



**Time Spent Reviewing:**

8 hours

---

> ### Author Response · Authors · 2021-08-10
> **Author Response to Reviewer ZmMv (1/2)**
>
> Thank you so much for your detailed, constructive, and useful comments; they will help us greatly improve the submission. We are also glad that you find our paper "clear and simple to follow" and our "physics-based policy directly guiding the training of kinematics-based model novel".  To address your comments and concerns:
>
> ---
>
> **Evaluation of the universal controller**
>
> We actually have evaluated motion imitation on **full AMASS dataset** but since it is used as training data, we only reported the test performance on Human3.6M. Here are the results on  the **full AMASS dataset** (containing 11299 motion sequences after data cleaning, as specified in Appendix C) using the controller reported in the paper:
>
> *Evaluation of motion imitation for our controller using target motion from the AMASS dataset.*
>
> |       AMASS Dataset | | | | |
> | ----------- | ----------- | ----------- | ----------- | ----------- |
> | Method  | $\text{S}_{\text{inter}} \uparrow$ | $\text{E}_{\text{root}}\downarrow$ | $\text{E}_{\text{mpjpe}} \downarrow$  | $\text{E}_{\text{acc}} \downarrow$
> | DeepMimic |  24.011%  | 0.385  | 61.634  | 17.938  |
> | UHC w/o MCP | 95.044%  | 0.134  | 25.254  |  5.383  |
> | UHC | **96.964%**  | **0.133**  | **24.454**  | **4.460**  |
>
> As can be seen from the table, our controller reported in the paper is able to perform 10956/11299 (96.964%) sequences from the full AMASS dataset, compared to the 2713/11299 (24.010%) sequences using DeepMimic. Upon visual inspection, our controller can imitate highly dynamic motion sequences such as dancing and kickboxing. Failure cases include some of the more challenging sequences such as breakdancing and cartwheeling. We will include the quantitative results and add additional visualizations of evaluation on the AMASS dataset. Our controller was able to achieve high imitation performance due to 1) better state and action design inspired by [1]; 2) incorporating residual force control (RFC) [2], the same technique utilized in SimPOE [3]. As argued in RFC [2], applying a residual force to the humanoid helps to compensate for the dynamics mismatch between the humanoid and a real human. Since the residual force is only applied at the humanoid's root, it does not affect the physical realism of human-object interaction and can produce physically realistic motion imitation (though this can be subjective). Our physics-based metrics such as foot sliding and penetration also indicate the physical plausibility of our controller. *We have subsequently developed a controller that only learns from a small, intelligently selected subset (243 sequences) of AMASS but is able to generalize to the full AMASS dataset ($\text{S}_{\text{inter}} = 99.531$%); we are happy to include the results if the reviewer requests it during rolling discussion*.
>
> We empirically verify that our controller can generalize to noisy estimates by using our controller on the result of a kinematic model trained **without** dynamics-regulated training (same setting as Row 1 of Table 2 in the main submission): our controller can mimic 143/183 (78.142%) estimated motion sequences without falling. This shows that our controller adapts to noisy estimates that are **physically invalid** (these sequences manifest large foot sliding and penetration, as indicated by Table 2) with a high success rate without additional training. Notice $S_{\text{inter}} $ reported in Table 2 is lower than 78.142% since $S_{\text{inter}} $ also factors whether the human-object interaction is successful.
>
> While the universal humanoid controller itself can be used for vision-based applications as a post-processing step, our main innovation also lies in the integration of a pre-trained controller in learning a kinematic policy. The pre-trained controller regulates the kinematic policy and provides a strong prior about physical plausibility. Thus, our dynamics-regulated kinematic policy provides a viable and efficient approach to learn a physics-based model using a standard MoCap dataset and can be deployed directly inside a physics simulator.
>
> ---
>
> **Insights into data and performance**
>
> We agree that the ill-posed nature of egocentric pose estimation makes it extremely difficult and a few assumptions were made on the captured data. We assume that we know the object of interest (to estimate its position and orientation), its class, and potential human-object interactions. As discussed in limitations, this constrains us to only estimate human-object interactions that we have data to learn from. Though our proposed dataset is not of the same scale as some of the third-person pose estimation datasets, our collected motion is diverse in trajectory and speed as discussed in Appendix D. For the actions that we support (sit, push, step, avoid, and locomotion), most do not manifest large arm movement (except for push), so their effect on MPJPE is small. Here we report the per-joint positional errors for the four joints with the smallest and largest errors, in ascending order:
>
> |       MoCap Dataset | | | | | | | |
> | ----------- | ----------- | ----------- | ----------- | ----------- | ----------- | ----------- | ----------- |
> | Torso | Left_hip  | Right_hip   | Spine   |  Left_toe | Right_toe | Right_hand | Left_hand |
> | 7.099 | 8.064 | 8.380 | 15.167 | 65.060 | 66.765 | 74.599 | 77.669 |
>
>
> As can be seen in the results, the toes and hands have much larger errors. This is expected as inferring hand and toe movement from only the egocentric view is challenging, and our network is able to extrapolate their position based on physical laws and prior knowledge of the scene context. Different from a third-person pose estimation setting, correctly estimating the torsor area can be much easier from an egocentric point of view since torso movement is highly correlated with head motion. In summary, the low MPJPE reported on our MoCap dataset is the result of 1) only modeling a subset of possible human action and human-object interactions, 2) the nature of the egocentric pose estimation task, 3) our network's incorporation of physical laws and scene context, which reduces the number of possible trajectories.
>
> This constrained setting still poses significant challenges. During the capture of our MoCap dataset, we asked the actors to vary their facing, gait, and speed. As a result, our MoCap sequences have an average speed range of [0.442, 1.029] meters per second and have many different approaching paths. The quantitative result in Table 1 in the manuscript shows that our method can adapt to different speed of motion sequences with a high success rate (96.9%), meaning that our simulated character stays stable and completes the human-object interaction without failure. Our captured real-world dataset also has more diverse and erratic motions (speed range of [0.227, 1.092]). Movement patterns such as zigzagging, circling, and side-stepping in the supplementary video (timestamp 03:41, 01:20, and 04:21) are **never** recorded in the MoCap dataset, and our model is able to produce reasonable motions. The high success rate of human-object-interaction also corroborates this observation. While the previous state-of-the-art methods struggle to produce stable human motion, our method is able to infer plausible poses and complete human-object interactions. As can be seen in the sitting, stepping, and avoiding sequences, people can have very different ways of turning around, sitting down, raising their feet, and avoiding. While previous state-of-the-art works [4] have been dedicated to synthesizing the sitting motion alone, our model can accurately predict stable sitting motion grounded in trajectories specified by egocentric videos. These results show that our model, only learning from a MoCap dataset (202 sequences), is able to generalize to real-world captures (183 sequences) without any additional fine-tuning. We will add more example sequences to the supplementary video to showcase our models' ability to generalize to different movement patterns.
>
> ---
>
> **Generalization**
>
> The objects **are not** positioned in the same location for the datasets: in *Figure 3 of the submitted appendix D*, we recenter the trajectories by the object positions to show the diversity of our movement patterns, as indicated by the caption. The real-world dataset is recorded in two different locations across multiple different recording sessions (e.g. the sequence at timestamp 0:55 is from an apartment common area, while 03:23 is at a living room). While the global poses of the objects can be drastically different, we use an agent-centric transform $T_{\text{AC}}$ to convert the detected object pose into the agent's coordinate frame, so our model can generalize to different spatial arrangements of objects. We will further discuss our object position diversity in the updated manuscript.
>
> In terms of the facing of the agent at the starting point of the sequence, we do collect some sequences where the actor is **not** facing the objects at the beginning of the sequence (e.g. at 01:48). This can also be seen in Figure 3  where some of the trajectories take a sharp turn in the beginning. As shown in the sitting motion and some sequences from the real-world dataset ( 03:40, 01:20, and 0:48), our model can execute heading change with high accuracy (the sequence at 01:20 records a full circling of the obstacle, which has never been recorded in the MoCap training data). Thus, given different facing directions at the start of the sequence, our model will be able to turn accordingly.
>
> We will further clarify the object position and orientation in the updated manuscript in Appendix D. Thank you for the advice!

---

> > ### Author Response · Authors · 2021-08-10
> > **Author Response to Reviewer ZmMv (2/2)**
> >
> > ---
> >
> > **Comments and technical questions**
> >
> > > How often is this distribution recomputed?
> >
> > The sampling distribution is calculated every 200 training episodes, which roughly corresponds to every 4 hours. We pre-compute the states needed for this procedure (which is a tensor of size 3661532x640), and it takes 35 seconds to use our value function to evaluate these states and recompute the sampling distribution. The $\widehat{\boldsymbol q_j }$ is indeed a typo; it should have been $\widehat{\boldsymbol q }_{j+1}$. Thank you for pointing it out!
> >
> > > Redundancy of information in the state space of the universal controller:
> >
> > By having redundant information we mean we have both joint angle and joint position in our state space, which contain a similar set of information (joint position can be calculated from joint angles using forward kinematics). This redundant information has been used as a form of regularization in prior and concurrent works [1, 5]. We have tried different configurations of the state space (where we added the center of mass differences as well), but have not seen noticeable differences in the performance of the controller. This is potentially due to the diversity and size of the AMASS dataset. We will further investigate the best set of state spaces for the universal humanoid controller.
> >
> > > Do the authors employ any technique, such as early stopping…?
> >
> > We utilize the same early stopping technique in DeepMimic and residual force control [2]. We will add this and other missing details to the updated manuscript.
> >
> > > The notation in Appendix C Policy Network architecture seems to be inconsistent.
> >
> > We will update the manuscript to correct these issues. Here we use $\ominus$ to indicate differences computed in quaternions. For rotation computed in extrinsic Euler angles, we directly use subtraction to compute their differences.
> >
> > > Will the authors release the dataset in case of publication to foster further research?
> >
> > All of our datasets, models, and code will be released for future research.
> >
> > > A potential missed citation
> >
> > Thank you for pointing it out! We will add this related work to our citation.
> >
> > ---
> >
> > **References:**
> >
> > [1] Chentanez, Nuttapong, Matthias Müller, Miles Macklin, Viktor Makoviychuk, and Stefan Jeschke. 2018. “Physics-Based Motion Capture Imitation with Deep Reinforcement Learning.” Proceedings - MIG 2018: ACM SIGGRAPH Conference on Motion, Interaction, and Games.
> >
> > [2] Ye Yuan and Kris Kitani. Residual force control for agile human behavior imitation and extended motion synthesis. In Advances in Neural Information Processing Systems, 2020.
> >
> > [3] Ye Yuan, Shih-En Wei, Tomas Simon, Kris Kitani, and Jason Saragih, “Simpoe: Simulated character control for 3d human pose estimation”, CVPR, 2021
> >
> > [4] Chao, Yu-Wei, Jimei Yang, Weifeng Chen and Jia Deng. “Learning to Sit: Synthesizing Human-Chair Interactions via Hierarchical Control.” AAAI (2021).
> >
> > [5] Rempe, Davis, Tolga Birdal, Aaron Hertzmann, Jimei Yang, Srinath Sridhar, and Leonidas J. Guibas. 2021. “HuMoR: 3D Human Motion Model for Robust Pose Estimation.” ICCV 2021
> >
> > [6] Peng, Xue Bin, Pieter Abbeel, Sergey Levine, and Michiel van de Panne. 2018. “DeepMimic.” ACM Transactions on Graphics 37 (4): 1–14.

---

> > > ### Comment · Reviewer_ZmMv · 2021-08-31
> > > **Thanks for the response**
> > >
> > > I appreciate the author's extensive answer to my concerns and questions. In the light of how the concerns were addressed, I'm willing to raise my score to 6. What I would like to see in the final version of the paper in case of acceptance is 1) a thorough discussion of the limitations (which is missing in the current main part of the manuscript) 2) the requested evaluations of the universal controller and the dataset statistics. 3) Clarification of how the pose metrics were obtained with respect to failing sequences. It seems that such good numbers for the pose can only be achieved if the sequences are ended for episodes deemed unsuccessful.
> > >
> > > I'm aware that this may be a bit late for the discussion, but since the authors mention that they have an improved controller that can be trained on a massively reduced subset of AMASS, I'm curious to know what insight this brings to training such a controller - both from the perspective of what data is most valuable for the learning process and what is implies with regards to the learned policy. It seems that the PD-controllers provide an  already reasonable initialization for achieving the pose and the policy mostly focuses on the residual force for keeping the character from falling. If the authors still find time to reply, it would be appreciated if they share their insights on this.

---

> > > > ### Author Response · Authors · 2021-09-01
> > > > **Discussion about improved controller**
> > > >
> > > > Thank you so much for reading our response and updating the review. We are glad and grateful that our response has addressed some of the concerns. We will continue to update our manuscript to add and clarify the important points raised during discussion (limitations, evaluation on AMASS dataset and additional data statics, etc.). Thanks again for your advice, which will greatly help improve the paper.
> > > >
> > > > Concerning the much-improved controller training on only a subset of sequences: we are actively working on this topic and are still formulating and testing our hypothesis. Our current hypothesis involves 1) the intelligently selected subset of "hard" poses guides the controller to prioritize stability over exact pose matching (though it still achieves good accuracy), 2) learning only on these "hard" poses help the agent better utilize residual force, which helps with long-term stability (though the residual force's scale stays the same in both cases). This is an ongoing and exciting project, and we would love to continue to share our findings with the community.

---

### Decision · Program_Chairs · 2021-09-27

**Decision:**

Accept (Poster)

**Comment:**

Originally there was a slight disagreement among the reviewers, with this paper receiving 3 positive ratings and 1 negative one. The reviewers acknowledge the interest of the studied scenario and that the empirical results are convincing. They nonetheless proposed several ways to improve the paper, by clarifying several aspects of the method and of the results, as well as discussing some additional works. The authors addressed these points in their feedback and managed to convince the most negative reviewer to raise their score. We therefore believe that this paper can be accepted to NeurIPS but strongly encourage the authors to incorporate their feedback in the paper for the final version.